

# Measurements of Delays of Gas-Phase Compounds in a Wide Variety of Tubing Materials due to Gas-Wall Interactions

Benjamin Deming,[1] Demetrios Pagonis,[1] Xiaoxi Liu,[1] Douglas Day,[1] Ranajit Talukdar,[1] Jordan Krechmer,[2] Joost A. de Gouw,[1] Jose L. Jimenez,[1] and Paul J. Ziemann[1]

[1] *Dept of Chemistry and Cooperative Institute for Research in Environmental Sciences, University of Colorado, Boulder, CO, USA*
[2] *Center for Aerosol and Cloud Chemistry, Aerodyne Research Inc., Billerica, MA, USA*
Correspondence to: Paul J. Ziemann (paul.ziemann@colorado.edu) and Jose L. Jimenez (jose.jimenez@colorado.edu)

## Abstract

Losses of gas-phase compounds or delays on their transfer through tubing are important for atmospheric measurements and also provide a method to characterize and quantify gas-surface interactions. Here we expand recent results by comparing different types of Teflon and other polymer tubing, as well as glass, uncoated and coated stainless steel and aluminium, and other tubing materials by measuring the response to step increases and decreases in organic compound concentrations. All polymeric tubings showed absorptive partitioning behaviour with no dependence on humidity or concentration, with PFA Teflon tubing performing best in our tests. Glass and uncoated and coated metal tubing showed very different phenomenology due to adsorptive partitioning to a finite number of surface sites. Strong dependencies on compound concentration, mixture composition, functional groups, humidity, and memory effects were observed for glass and uncoated and coated metals, which (except for Silonite-coated stainless steel) also always caused longer delays than Teflon for the compounds and concentrations tested. Delays for glass and uncoated and coated metal tubing were exacerbated at low relative humidity but reduced for RH > 20%. We find that conductive PFA and Silonite tubing perform best among the materials tested for gas plus particle sampling lines, combining reduced gas-phase delays with good particle transmission.



# 1 Introduction

A number of studies have demonstrated that absorptive partitioning of volatile organic compounds (VOCs) into the Teflon walls of environmental chambers can affect the results of the experiments
conducted within. This partitioning has been shown to be reversible and relatively fast, on a timescale of minutes (Matsunaga and Ziemann, 2010; Yeh and Ziemann, 2015; Krechmer et al., 2016; Ye et al., 2016; Huang et al., 2018). Furthermore, a recent study showed analogous absorptive partitioning of VOCs when transported through PFA (perfluoroalkoxy alkanes) Teflon tubing, with similar interaction parameters as for FEP (fluorinated ethylene propylene) Teflon chamber walls (Pagonis et al., 2017).
That work found that the tubing acted roughly as a chromatography column, effectively smearing the time profile of the measured compounds and affecting the measured concentrations. Delays of over 10 minutes were observed for realistic conditions for the least-volatile compounds ($C^* \sim 3$ x $10^4$ µg m$^{-3}$) with longer delays predicted for compounds less volatile than those measured in that study. The resulting time profiles were well-reproduced by a simple numerical chromatography model that divided
the length of tubing into a series of bins in which organic compounds partitioned between the gas phase and the walls based on the vapor pressure of the organic compound and an equivalent absorbing mass of the wall ($C_w$, µg m$^{-3}$) according to Eq. (1):

$$F_w = \frac{1}{1+\frac{C^*}{C_w}}$$
(1)

In this equation, $F_w$ is the fraction of the compound partitioning to the wall at equilibrium, and $C^*$ (µg
m$^{-3}$) is the saturation concentration (the vapor pressure in mass units) of the organic compound estimated using the SIMPOL.1 group contribution method (Pankow and Asher, 2008). The model code was made publicly available in the paper. They also demonstrated that partitioning depended only on the saturation concentration of the organic compound and not its specific functionality.

Although PFA Teflon is one of the most commonly used materials for gas sampling lines and
instrumentation surfaces, a wide variety of materials finds use in practice for sampling gases, including other types of Teflon, PEEK (polyether ether ketone), glass, and uncoated and coated stainless steel and aluminum. Quantitative aerosol sampling requires electrically conductive tubing to avoid major losses



of charged particles in Teflon tubing, and is commonly performed using uncoated stainless steel,
copper, aluminium, or polymeric tubing that has been rendered conductive by additives such as black

carbon. Partitioning of semi-volatile gases to and from tubing and instrument internals can disturb
gas/particle equilibrium, resulting in additional evaporation or condensation of material that may
interfere with measurements. In oxidation flow reactors, such tubing and inlet delays can perturb the
equilibrium of lower volatility compounds that are thought to dominate potential aerosol mass (Palm et
al, 2018). Decisions on material choice are based on a number of criteria, including but not limited to

cost, weight, and electrical conductivity (for aerosols only).  Also considered are the potential for gas-
phase losses, delays, measurement artifacts, or memory effects, particularly when measuring lower
volatility compounds or those that interact strongly with surfaces. However, systematic testing of the
effects of different materials on measurements has been limited.

Here we present results of a systematic survey of 14 commonly-used tubing materials with the

same compounds, conditions, and measurement protocol. The effect of tubing on measurements was
characterized by introducing step-function changes in compound concentrations while sampling through
a length of tubing or directly into the instrument inlet, allowing characteristics of the tubing to be
separated from any instrument effects. Through these measurements, the physical basis of partitioning
in different materials can be understood, and relative performance of the different materials can be

compared accurately. We aim to facilitate more informed decisions about material choice for sampling
lines and inlet and instrument design, and also to provide information on gas/surface interactions that
may be useful to interpret studies in indoor air chemistry and other fields.

## 2 Experimental

### 2.1 Absorbent tubing experiments

A series of experiments was conducted with various polymeric tubing materials that are listed in Table
1. Selected 2-ketones and 1-dodecene were added to an 8 m$^3$ FEP Teflon environmental chamber (the
"VOC chamber"), which was filled with purified air from an AADCO 737 Pure Air Generator. 2-
Hexanone (99%), 2-octanone (98%), 2-decanone (98%), 2- tridecanone (99%), and 1-dodecene (95%)



Table 1. Tubing materials investigated in this study.

| Material | Classification | Internal Diameter (cm) | Supplier (Part No.) |
|---|---|---|---|
| PFA (perfluoroalkoxy alkanes) | Absorbent | 0.476 | McMaster-Carr (52705K34) |
| FEP (fluorinated ethylene propylene) | Absorbent | 0.476 | McMaster-Carr (2129T13) |
| PEEK (polyether ether ketone) | Absorbent | 0.381 | BGB Analytik |
| PTFE (polytetrafluoroethylene) | Absorbent | 0.476 | McMaster-Carr (5239K12) |
| C-PTFE (conductive PTFE) | Absorbent | 0.476 | Finemech (S1827-68) |
| C-PFA (conductive PFA) | Absorbent | 0.476 | Fluorotherm |
| Aluminum | Adsorbent | 0.457 | McMaster-Carr (89965K431) |
| Chromated aluminium | Adsorbent | 0.457 | As above, then chromated by KMG Industrial Screening & Metal Finishing, Inc. |
| Electropolished steel | Adsorbent | 0.457 | Harrington Pure |
| Copper | Adsorbent | 0.483 | Grainger (2LKK2) |
| Glass | Adsorbent | 0.457 | CU glassblowing workshop |
| Silcosteel | Adsorbent | 0.457 | Restek |
| Stainless steel | Adsorbent | 0.457 | McMaster-Carr (89895K724) |
| Silonite | Adsorbent | 0.457 | Entech Instruments |

were obtained from Aldrich; and 2-dodecanone (98%) and 2-tetradecanone (98%) were obtained from
ChemSampCo. Solid standard compounds were weighed and added to a glass bulb, whereas liquids
were measured via syringe and dispensed directly into the same bulb. These standards were then
evaporated and flushed from the bulb (with heating in some cases) directly into the chamber using a 5 L
min$^{-1}$ stream of ultra-high purity (UHP) N$_2$ (Airgas). The initial concentration in the chamber was





approximately 20 ppb for each compound prior to gas-wall partitioning. Using Eq. (1), $C*$ values estimated using SIMPOL.1 (Pankow and Asher, 2008), and a $C_w$ value of 20 mg m$^{-3}$ (Matsunaga and Ziemann, 2010; Yeh and Ziemann, 2015), chamber concentrations ranged from 20 ppb for the most

volatile compound (2-hexanone) to approximately 13 ppb for the least volatile (2-tetradecanone). It should be noted, however, that since all signals are normalized to the measured chamber signal the absolute concentrations do not matter for this analysis. For experiments conducted under dry conditions the humidity was less than 0.5% RH, whereas for humid experiments the desired RH was achieved by adding HPLC-grade water to the chamber in the same manner as described above for the VOCs. An

FEP Teflon-coated fan was run for ~1 min to guarantee complete mixing and to help achieve gas-wall partitioning equilibration in the 30 min period before measurements were taken. A second chamber (the "clean chamber") contained only purified air, and in some cases added water vapor. The chambers operated at room temperature (23ºC +/– 2ºC), which was typically stable within 1ºC, and the humidity (measured using an Amprobe THWD-5) of the two chambers differed by less than 5% RH.

After the chambers had equilibrated, the instrument inlet was connected to the VOC chamber via the tubing to be investigated. The flow rate through the inlet was maintained at 0.300 ± 0.015 L min$^{-1}$ for all experiments. Once the measured signals had reached steady-state, meaning that both the tubing and the instrument were equilibrated with the gas phase ("passivation"), sampling was rapidly switched to the clean chamber either before the tubing entrance (to measure the total delay due to instrument and

tubing) or directly at the instrument inlet (to measure the instrument delay only) ("depassivation").

Delays were quantified by fitting the measured depassivation time series to exponential decays. The tubing delay for these experiments is defined in Eq. (2) as the difference in the time it takes each of these curves to reach 90% of the final value:

$$t_{tubing,abs} = \ln(10)\,(\tau_{total} - \tau_{instrument}) \qquad (2)$$

where $t_{tubing,abs}$ is the absorptive tubing delay, $\tau_{total}$ is the fitted timescale for the tubing plus instrument depassivation, $\tau_{instrument}$ is the fitted timescale from the instrument-only depassivation, and the factor of $\ln(10) = 2.3$ accounts for the difference between the fitted timescales and the time required to reach 90% depassivation. Comparing these two depassivation timescales allows the tubing delay to be





decoupled from the instrument response. Each tubing delay was then normalized by the length of the
piece of tubing used. Note that we use $t$ to refer to measurement delay times and $\tau$ to refer to fitted
exponential timescales. A derivation of this equation can be found in the Supplement. The tubing model
of Pagonis et al. (2017) was used to simulate the tubing delays expected for different values of $C_w$
across the range of $C^*$ of the compounds investigated. The value of $C_w$ resulting in the lowest error
(calculated as the sum of squared residuals between modelled and measured delay curves) was chosen
to be the best estimate.

## 2.2 Adsorbent tubing experiments

A series of experiments was conducted with various uncoated and coated metal and glass tubing that are
listed in Table 1. To avoid surface displacement processes that can occur with these materials
(discussed in detail further below), only a single compound (2-hexanone, 2-decanone, or 1-dodecene) at
a time was loaded into the chamber for most experiments. Each sample of tubing was depassivated
using air from the clean chamber until steady-state values were reached. The tubing was then connected
directly to the VOC chamber and sampled until a steady-state signal was reached. Because the time
series consisted of a long period with no signal followed by an approximately sigmoidal increase in
signal, the tubing delay for this adsorptive tubing, $t_{tubing,ads}$, is defined as the time it takes the measured
signal to reach 50% of its steady-state value during passivation. This value was determined by selecting
the points between 35% and 65% of the maximum, performing a linear fit, and solving the linear fit
equation for the point at which 50% was reached. These delays were then corrected for the measured
instrument response. Because $t_{total}$ is defined differently than $\tau_{total}$, adsorptive tubing delays were
calculated using Eq. 3:

$$t_{tubing,ads} = t_{total} - \ln(10)\,\tau_{instrument} \qquad (3)$$

where $t_{tubing}$ is the tubing delay, $t_{total}$ is the measured passivation delay (calculated as described in
Section 2.1), and $\tau_{instrument}$ is the measured instrument timescale for the compound. A derivation of this
equation is given in the Supplement and an example time series is shown in Fig. S1.





## 2.3 Instrumentation

The quadrupole proton transfer reaction-mass spectrometer (q-PTR-MS) used in this work has been previously described (de Gouw, 2007). The inlet system was reduced prior to these experiments by removing a length of Silcosteel tubing (~1 m, 1/8" OD) and simplifying the valve system. Experiments were performed after the instrument had been pumped down and running for several days. A Vocus proton-transfer reaction-time of flight mass spectrometer (Vocus PTR-TOF) was also used for several
experiments (Krechmer et al., 2018).

## 3 Results and Discussion

### 3.1 Independent absorptive versus competitive adsorptive behavior

In surveying different tubing materials it became evident that two fundamentally different mechanisms for passivation/depassivation exist. Example time series for the passivation and depassivation of 3 m of
FEP Teflon and 1 m of stainless steel tubing are shown in Fig. 1. Although the experimental procedures were identical, the resulting time series for the FEP Teflon (Fig. 1a and b) and stainless steel tubing (Fig. 1c and d) show significant differences that give insight into the sorption mechanisms responsible for the tubing delays. For the FEP Teflon tubing, the approximately exponential build-in of signal during passivation (Fig. 1a) and decrease during depassivation (Fig. 1b) are consistent with an
absorptive process in which each compound partitions into the tubing walls according to its vapor pressure and independent of interactions with the other compounds (Pagonis et al., 2017). This contrasts with the behavior seen for the stainless steel tubing (Fig. 1c), in which there is a period of nearly an hour before any signal is measured, followed by a transient enhancement in the signal of the most volatile compound (2-octanone) above that corresponding to its concentration in the chamber (measured
separately the same day using FEP Teflon tubing as an inlet). This transient enhancement ends as the signal from the next most volatile compound in the homologous series (2-decanone) grows in, and then after that signal peaks the same pattern of signals occurs sequentially for the other 2-ketones in the series. This behavior suggests that during the period when no signal is measured the compounds in the mixture are all adsorbing to unoccupied surface adsorption sites, and that once these sites are all filled
the compounds competitively displace one another according to their vapour pressures as they travel





Figure 1. (a) Passivation and (b) depassivation curves measured for step function changes in a series of
2-ketones sampled through absorbent tubing (3 m of FEP Teflon). (c) Passivation and (d) depassivation
curves measured for step function changes in a series of 2-ketones sampled through adsorbent tubing (1
m of stainless steel). Absorbent and adsorbent tubing was depassivated using dry and 40% RH air,
respectively. Note the different scales in panels c and d.

through the tubing. When an identical experiment was conducted with a single ketone no enhancement

in the concentration above the chamber concentration was observed (as in Fig. S1), which we take as



further evidence for competitive adsorption in the mixture experiment. This conclusion is also based on the characteristics of the time series presented in Fig. 1d, which were measured when the passivated stainless-steel tube was depassivated with humid room air. Rather than appearing as a series of

exponential decays (as seen for FEP Teflon in Fig. 1b), the measured concentrations are again enhanced above the chamber concentration (by up to a factor of 40) before approaching zero (the background level for room air). In this experiment, 2-ketones adsorbed to the stainless steel are suddenly displaced by water, causing rapid desorption that leads to the enhancement in measured compound concentrations. We therefore use the observation of a strong humidity dependence in measured tubing delays as

additional evidence that the delays are controlled by adsorption and use it as an identifying characteristic to divide the investigated materials into two classes (Table 1): absorptive (independent VOC absorption, RH-independent, polymer-like) and adsorptive (competitive VOC adsorption, RH-dependent, metal-like).

The conclusion that there are two sorption mechanisms at play is supported by measurements of

partitioning of VOCs to Teflon membrane filters and quartz filters by Mader and Pankow (2000, 2001). Although these authors framed their findings as adsorption in both cases, they report that partitioning to Teflon showed no humidity dependence and was not influenced by other compounds in the ambient air sampled (Mader and Pankow, 2000). In contrast, sorption to the quartz filters was strongly humidity dependent and influenced by other organic compounds (Mader and Pankow, 2001), consistent with our

hypothesis that sorption to some polymeric materials occurs independent of intermolecular interactions by absorption while for some other materials it occurs by competitive adsorption. Further evidence of competitive VOC adsorption also appears in the work of Roscioli et al. (2015). These authors found that active, continuous passivation of the glass inlet and internal surfaces of an instrument with surface-active fluorinated acidic or basic compounds improved the time response for nitric acid and ammonia,

respectively. Upon initial passivation they observed spikes in nitric acid or ammonia concentrations (similar to the behaviour in Fig. 1d) that corresponded to displacement from surfaces.





## 3.2 Measurements of absorptive delays

The measured tubing delays of 2-ketones through polymer-like, absorbent materials (PFA, FEP, PTFE,
PEEK, and conductive PTFE) under dry conditions are shown in Fig. 2. The lines are model runs fitted
to the experimental data, which reproduce the observed trends well and were used to calculate the $C_w$
values for each tubing material given in Table 2. These $C_w$ values may be used in conjunction with the
model presented by Pagonis et al. (2017) to simulate the effects of different sampling lines on measured
gas-phase concentrations. When applied to tubes with other diameters or to these materials in other
geometries $C_w$ should be scaled by the surface-to-air volume ratio (Pagonis et al., 2017). PFA Teflon

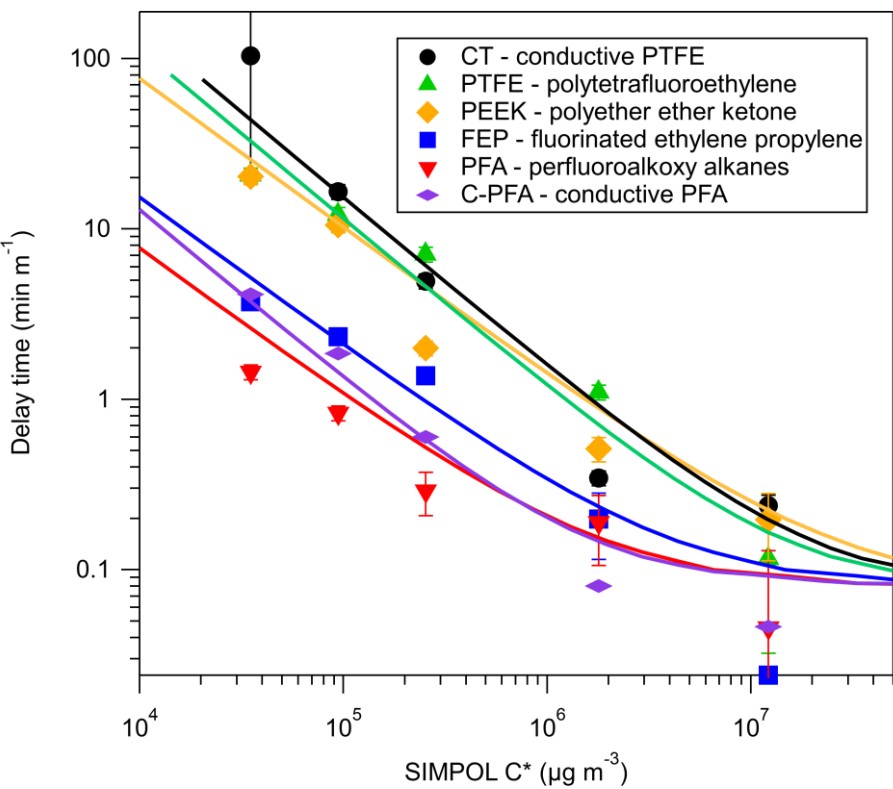

Figure 2. Measured tubing delays normalized by tubing length as a function of the saturation
concentration of a series of 2-ketones as estimated by SIMPOL.1. Error bars were propagated from
exponential fits of depassivation curves.   Lines are results from the Pagonis et al. (2017)
chromatography model.





Table 2. Fitted values of $C_w$ for absorbent tubing materials.

| Tubing Material | $C_w$ (µg m⁻³)[a] | Internal Diameter (cm)[b] | Internal Surface Area/ Volume Ratio (cm⁻¹) |
|---|---|---|---|
| PFA (perfluoroalkoxy alkanes) | $8.0 \times 10^5$ | 0.476 | 8.40 |
| C-PFA (conductive  PFA) | $1.3 \times 10^6$ | 0.476 | 8.40 |
| FEP (fluorinated ethylene propylene) | $2.0 \times 10^6$ | 0.476 | 8.40 |
| PEEK (polyether ether ketone) | $8.0 \times 10^6$ | 0.381 | 10.5 |
| PTFE (polytetrafluoroethylene) | $1.2 \times 10^7$ | 0.476 | 8.40 |
| C-PTFE (conductive PTFE) | $1.6 \times 10^7$ | 0.476 | 8.40 |

[a]Values of $C_w$ for other conditions should be scaled proportionally to the surface-to-volume ratio.
[b]Outer diameter = 0.635 cm.

appears to outperform FEP Teflon in terms of measurement delays, although the differences may be within our estimated error and within the level of reproducibility observed for different pieces of tubing
of the same material. PEEK, PTFE, and conductive PTFE showed significantly larger delays than PFA and FEP Teflon. According to Fluorotherm (2018), PFA and FEP both have shorter polymer chain lengths and increased chain entanglements as compared to PTFE. Absorption into Teflon likely occurs as gas molecules fit into spaces between the polymer chains as they thermally oscillate (Yeh et al., 2015). The fact that the polymer chains of PTFE are not as tangled as those of PFA and FEP suggests
that there may be more spaces available for gases to absorb into the material, consistent with the larger value of $C_w$ determined here. Conductive Teflon is PTFE or PFA with added black carbon to make the tubing electrically conductive and therefore appropriate for sampling aerosol. It does not appear that the added black carbon significantly changes the partitioning properties of the tubing for the compounds studied here. Transport of charged particles through conductive PFA tubing wrapped in aluminium foil
(to further prevent static build-up) was comparable to sampling through copper, as shown in Fig. S3.



Contrary to the commercially available conductive silicone tubing (Timko et al., 2009; Yu et al., 2009), we did not observe emission of any species from the tubing with either the VOCUS PTRMS in this study, or with an I⁻ CIMS in a related study (Liu et al., 2019). Since conductive PFA combines low interaction with gases with the electrical conductivity needed to sample particles it is an optimal choice

for applications that require joint gas/aerosol sampling lines. As mentioned previously, measurement delays due to absorbent, polymer-like tubing do not exhibit humidity dependence for the species studied here, as demonstrated for PFA, FEP, and PTFE in Fig. S4. We also note that we briefly investigated a short (0.60 m) length of Nafion tubing (0.178 cm ID) using the methodology described for absorbent tubing. After 30 min even the most volatile ketone (2-hexanone) had only reached 30% of the chamber

concentration (Fig S2), so the experiment was aborted. It thus appears that this tubing may interfere with the sampling of polar compounds, although further investigation is needed to determine the reason.

### 3.3 Measurements and characterization of adsorptive delays

A full, quantitative investigation into the adsorptive mechanism reported here is not attempted in this paper. Instead, we discuss a few general findings that we hope will help inform decisions in designing

sampling schemes. As discussed above, the measurement delays arising from adsorptive, or metal-like, tubing materials are highly humidity dependent. This is highlighted in Fig. 3, where an increase from 0% to 20% in RH generally decreases the tubing delay by about an order of magnitude. The longest measured delay times were for aluminum tubing and aluminum tubing treated with hexavalent chromate conversion coating. This treatment is intended to prevent corrosion and is used in the Potential Aerosol

Mass flow reactor (Kang et al. 2007), and it does not appear to significantly affect the measurement delays. Stainless steel, Silcosteel, copper, and glass showed similar results to each other, among the lowest delays for the adsorptive-type tubing. Surprisingly, electropolished steel performed worse than regular stainless steel. Electropolishing creates a very smooth surface, which should theoretically reduce the number of available surface sites for adsorption. It is possible that the polishing did reduce the

internal surface area of the tubing, but actually increased the number of sites by changing the elemental composition or microstructure of the surface. Alternatively, it should be noted that the lengths of steel and copper tubing used in these measurements had previously been used in laboratory experiments,





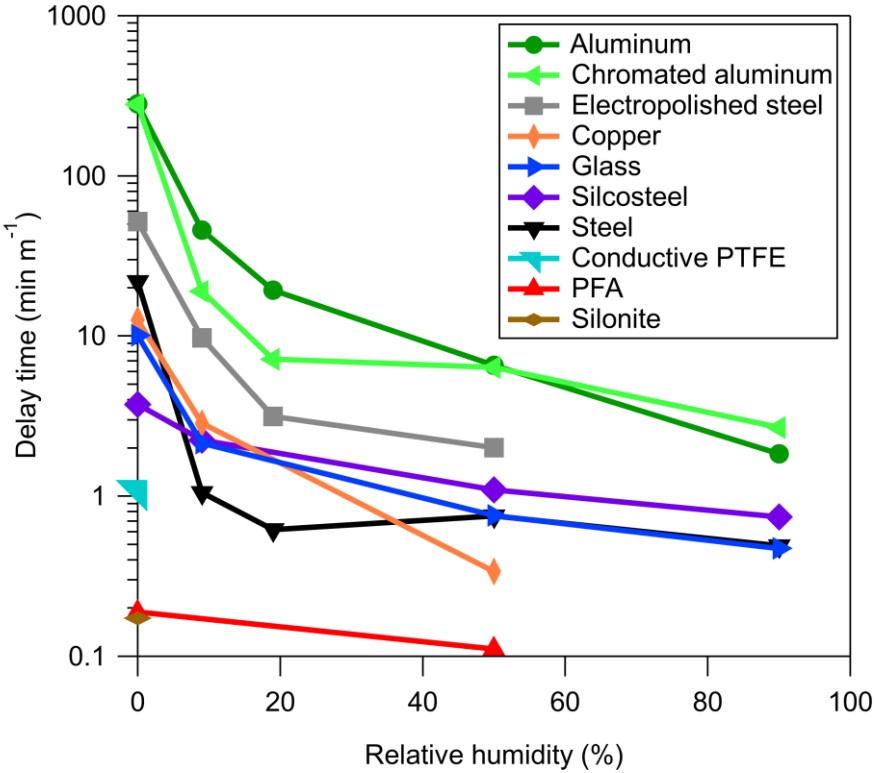

Figure 3. Humidity-dependent delay times measured for 2-decanone for a series of adsorptive tubing materials. Conductive PTFE and PFA are also included for comparison.

which included sampling compounds with lower saturation concentration than 2-decanone. It is possible that some of these compounds remained adsorbed to the tubing even after depassivation with clean air, effectively conditioning these tubing samples by blocking some of the adsorption sites. The other tubing samples, in contrast, were bought new and used only for the adsorptive delay experiments. This could partially explain the discrepancy between normal and electropolished stainless steel. Supporting this argument, the measurement delay through a short length of electropolished steel previously used for SOA sampling in hundreds of chamber experiments (and therefore coated in low volatility organic compounds) was also measured and found to be much lower than both stainless and electropolished steel.





Although we define the delay times slightly differently for the two types of tubing, the values for PFA Teflon and conductive Teflon are also shown in Fig. 3 for comparison. In addition to exhibiting less complex behaviour, the delays produced using PFA Teflon are shorter than the majority of the

tested adsorptive materials, even at high RH. Notably, although Silonite tubing performed as well as PFA Teflon in terms of delay, even at low relative humidity, and exhibited good particle transmission (Fig S3), measurements may be influenced by humidity and VOC-VOC interactions.

In addition to humidity, the measured tubing delay depends on the concentration of the compound being measured, as seen in Fig. 4. At a concentration of approximately 100 ppb of 2-

hexanone it appears that the stainless-steel tube is saturated: increasing the concentration no longer decreases the measured delay. Using the flow rate and internal surface area of the tube, this corresponds

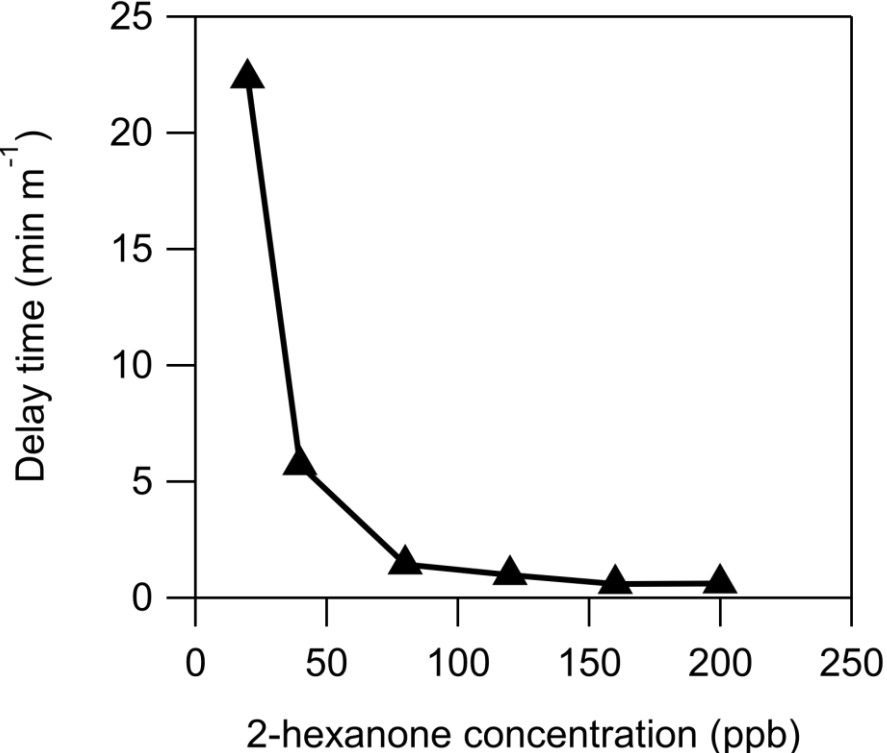

Figure 4. Measurement delays of 2-hexanone through 1 m of stainless steel as a function of chamber 2-hexanone concentration under dry (RH < 0.5%) conditions.



to an estimated coverage of 4 x $10^{13}$ molecules cm$^{-2}$. This is slightly less than the surface concentration

of 1.4 x $10^{14}$ molecules cm$^{-2}$ measured by Vaittinen et al. (2014) for ammonia on steel, which may be

due in part to the larger size of 2-hexanone molecules. Additionally, these authors had more control

over the humidity in their system than was possible in this work, and since even small changes in the

relative humidity can drastically affect the measured delay time it is possible that our value is artificially

low. Furthermore, because the adsorption mechanism appears to be competitive and this length of

tubing had been previously exposed to organics with much lower vapor pressure than 2-hexanone it is

possible that some of those organics remained sorbed to the tubing, effectively reducing the number of

adsorption sites available for 2-hexanone.

Partitioning in adsorbent tubing is also dependent on the functionality of the sorbed compound,

rather than solely on saturation concentration as was demonstrated to be the case for absorbent tubing,

at least for alkenes and ketones (Pagonis et al, 2017). Fig. 5 compares the delays measured for 2-

decanone and 1-dodecene: two compounds with similar saturation concentrations as estimated from

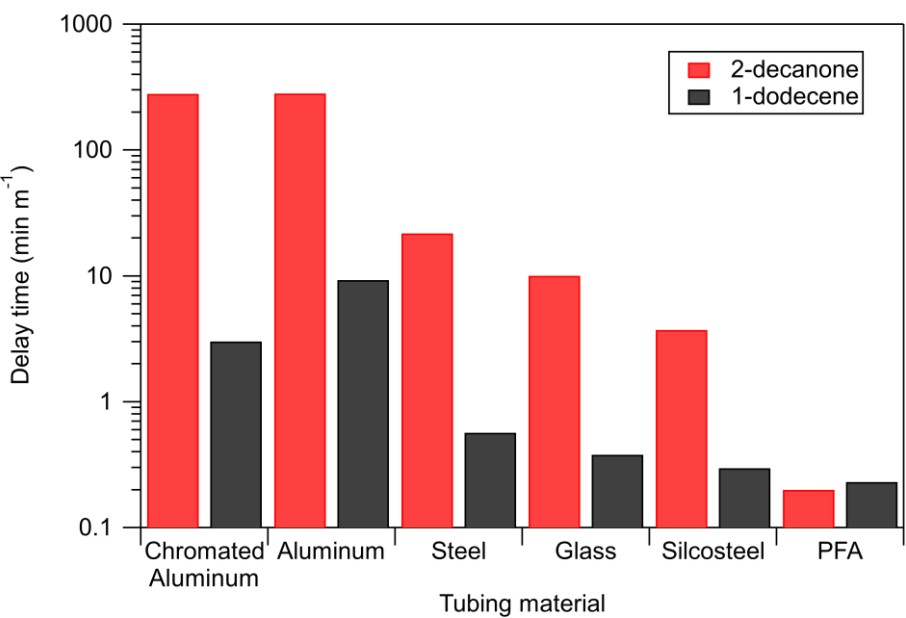

Figure 5. Comparison of measured tubing delay times for a ketone and an alkene of similar saturation
concentration (2-decanone and 1-dodecene). All measurements were performed under dry conditions
(RH < 0.5%) with 20 ppb of the standard in the VOC chamber.





SIMPOL.1 ($1.8 \times 10^6$ µg m$^{-3}$ and $1.6 \times 10^6$ µg m$^{-3}$) but different functionality. For all but the absorptive PFA tubing the delays are much shorter for the alkene than the ketone, suggesting that more polar compounds adsorb more strongly. This is consistent with our explanation of adsorption as the mechanism behind these delays, as adhesion to surface sites may depend on molecular weight, polarity, or even specific functionality. It is also consistent with the ability of water vapor to displace the ketones from the sites. It should be noted that the alkene delay measurements are more uncertain than the ketone measurements due to lower instrument sensitivity, but because the delay times typically differ by an order of magnitude or more we conclude that this functionality dependence is real.

**4**. **Conclusions**

Building on the work of Pagonis et al. (2017), we measured the tubing delays associated with sampling VOCs through a wide array of tubing materials. It was found that delays through polymer tubing are controlled by independent absorption, whereas delays in glass and uncoated and coated metal tubing are controlled by competitive adsorption. Absorbent tubing exhibits delays that can be characterized by an effective absorbing mass concentration of the wall, which we report here for six different materials, and which can be scaled for other tubing sizes or material geometries. These values can be used in the model provided by Pagonis et al. (2017) to predict the effects of sampling lines on measurements. Furthermore, delays in absorbent tubing do not show humidity, concentration, or functionality dependence over the ranges of these variables tested here. This is in contrast to adsorbent tubing, which demonstrates a strong dependence on these three factors in addition to generally longer delay times. We therefore recommend the use of absorbent tubing when possible to simplify analysis of gases. If they can be used, PFA and FEP Teflon appear to be the best choices for minimizing measurement delays. If adsorbent tubing must be used, delays can be minimized by ensuring the RH is maintained above 20%. It should be emphasized that use of adsorbent tubing can result in large memory effects and sampling artifacts, particularly upon changes in RH. Conductive PFA tubing and Silonite were shown to be the best choices for simultaneous gas and particle sampling; however we note that the Silonite purchased





here cost 2.5 times that of conductive PFA per foot. Despite these recommendations, adsorbent materials will no doubt continue to find use in sampling lines and instrument internal surfaces. Further

work is therefore necessary to more completely characterize the relationships put forth in this paper. Specifically, the effects of functionality and concentration should be analyzed more fully to develop a better working model for the mechanism of these measured delays.

### Acknowledgements

This work was supported by the Dept. of Energy (BER/ASR) DE-SC0016559 and by the Sloan Foundation program on the Chemistry of Indoor Environments (CIE, Grants 2016-7173 and 2018-10071). We thank the ToF-CIMS users community, Bill Brune, and Eric Apel for useful discussions.

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
