# Peer review of "Measurements of Delays of Gas-Phase Compounds in a Wide Variety of Tubing Materials due to Gas-Wall Interactions"

_Atmospheric Measurement Techniques, 2019_

## Referee Comment (RC1) · Anonymous Referee #1 · 26 Feb 2019

Reviewer comments on amt-2019-25.

The authors present a very useful comparison between many widely used tubing types. They report equilibration times of each tubing material when switching between sampling representative atmospheric compounds and clean air. In particular, polymeric tubing (i.e. Teflon) is reported to generally have substantially faster equilibration times than metal tubing (in most cases even when coated). While relatively technical in nature, this work tackles an important issue in designing new instrumentation for the analysis of difficult-to-measure atmospheric components. I have relatively few comments and recommend publication with only minor revisions.

[Figure]

Major comments: 1) It is not clear to me how well adsorbent and absorbent tubing can be compared by their approach, which is concerning given that a major conclusion of this work is the advantage of absorbing tubing over adsorbing tubing for sampling gases. One specific issue on this topic is that adsorbing times are on a 50% benchmark and the absorbing times on a 90% benchmark. Are these quantitatively comparable, which is to say, do they have the same mathematical meaning to allow direct comparison across modes? In Eq. (3), it seems odd to subtract the time it takes to get to 90% in the instrument from the time it takes to get to 50% in the complete setup. Couldn't the sigmoidal fit of the adsorbing data be used to similarly estimate time it takes to get to 90% and provide a uniform comparison? On a similar note, why where timescales measured during depassivation for absorbent and passivation for adsorbent?

The authors explicitly discuss that these metrics are different, so different equations are used and when they are included on the same plots a note is made. However, the authors nevertheless compare these cases, for instance stating in the abstract that "glass and uncoated and coated metals ... always caused longer delays than Teflon." I think it is important to compare across these materials, so my suggestion is not to stop comparing (e.g. removing the offending sentence in the abstract), but to put a little more care into figuring out how best to compare across categories (e.g. unify benchmarks).

2) A lot of time is spent rationalizing and discussing the fact that the steel tubing was previously used. I'm not sure how best to handle this; in short my issue is that large sections of pages 13 and 15 discuss the potential impact of this issue on the observed results but fundamentally it is just an N of 1. If the authors truly believe that the results of steel are strongly influenced by the history of the tubing, it seems best to just leave that data out and focus on 13 instead of 14 types of tubing. Otherwise, given the amount of time these seem to need to talk about it, it is apparently a bit of an apples to oranges comparison. Maybe they could have another small paragraph if which they discuss the possibility that tubing history has an impact and present their evidence for

that specific issue there.

Technical comments: I actually have very few minor technical comments. I noticed no typos or specific issues in language, and believe the figures are clear and to the point.

I notice that the manufacturers of C-PFA also makes C-FEP, and it seems to be significantly cheaper (a little more than half the price). Given the relatively similar results between FEP and PFA, is there a reason the authors chose to test C-PFA but not C-FEP? Convincing the community to switch to conductive Teflon would be easier if cheaper, so it is a little bit unfortunate that C-FEP was not tested or discussed.

I think Figure S3 actually adds a lot of insight, and should maybe be added to the main body of the manuscript. If this change were made, the figure would need some cleanup to bring it up to the clarity standards of the current main figures.
* * *

---

## Referee Comment (RC2) · Anonymous Referee #2 · 25 Mar 2019

General comments:

In analytical chemistry of gases and particles interactions with wall material of tubing, adsorbents, and/or material of other devices are often crucial. This paper investigates the impact of various tubing materials on the recovery of 4 ketones (C6-C13) and one C12-hydrocarbon in different humidity regimes. The authors present the results clearly and well structured.

Specific comments:

The gas/particle phase analytical community certainly appreciates such an investigation assisting in choosing an appropriate tuning material in the applications. The tubing material selected in the experiments is well tuned far from being comprehensive. It may be worthwhile to discuss more deeply effects of humidity and aging of tubing material on adsorption/desorption processes and thus related recovery of the target compounds. The paper is recommended for publication with minor revisions considering the technical comments.

Technical comments:

The author team should thoroughly check the comma settings throughout the text. This would lead to better structured text improving text understanding.

Page 2 line 47: . . . demonstrated that partitioning is depended . . .

Page 4/5 line 86/87: per definition ppb in not a concentration. It is a mixing ratio. It is not clear whether weight or volume of the compound of interest is used. It is recommended to use amount fraction or mol fraction throughout the text.

Page 5 line 101: please explain how the flow of 300 ml/min was adjusted. A possible artefact formation through the flow regulating device should be discussed.

Figure 1 legend: the dashed line represents not the chamber concentration. Please describe the correct meaning of the dashed line in the legend below the figure and remove dashed line and label from the legend box in Figure 1a.

Page 12 line 249: "... we hope will help inform decisions in designing . . .": The meaning is not clear. Please rephrase.

Page 12 line 253 to line 256: "The longest measured delay times were for aluminium tubing and aluminium tubing treated with hexavalent chromate conversion coating. This treatment is intended to prevent corrosion and is used in the Potential Aerosol Mass flow reactor (Kang et al. 2007), and it does not appear to significantly affect the measurement delays." Please check for consistency: "longest measured delay times" versus "does not appear to significantly affect the measurement delays". Correct "aluminum".

Figure 2, legend x-axis: what is the meaning of the "*" in SIMPOL C? Please explain or delete.

Page 14 line 283: Please start new sentence after "(Fig S3)". ... (Fig S3). The measurements may be . . ..

---

## Author Comment (AC1) · 8 May 2019

**Response to Reviewer Comments**

**Measurements of Delays of Gas-Phase Compounds in a Wide Variety of Tubing Materials due to Gas-Wall Interactions**

Benjamin Deming,[1] Demetrios Pagonis,[1] Xiaoxi Liu,[1] Doug Day,[1] Ranajit Talukdar,[1] Jordan Krechmer,[2] Joost A. de Gouw,[1] Jose L. Jimenez,[1] and Paul J. Ziemann[1]

[1] Dept of Chemistry and Cooperative Institute for Research in Environmental Sciences, University of Colorado, Boulder, CO, USA

[2] Center for Aerosol and Cloud Chemistry, Aerodyne Research Inc., Billerica, MA, USA

Correspondence to: Paul J. Ziemann (paul.ziemann@colorado.edu) and Jose L. Jimenez (jose.jimenez@colorado.edu)

We thank the reviewers for their insightful comments. They are copied and numbered below and each one is followed by our response.

**Reviewer 1**

*General Comments*

The authors present a very useful comparison between many widely used tubing types. They report equilibration times of each tubing material when switching between sampling representative atmospheric compounds and clean air. In particular, polymeric tubing (i.e. Teflon) is reported to generally have substantially faster equilibration times than metal tubing (in most cases even when coated). While relatively technical in nature, this work tackles an important issue in designing new instrumentation for the analysis of difficult-to-measure atmospheric components. I have relatively few comments and recommend publication with only minor revisions.

*Specific Comments*

Comment 1.  It is not clear to me how well adsorbent and absorbent tubing can be compared by their approach, which is concerning given that a major conclusion of this work is the advantage of absorbing tubing over adsorbing tubing for sampling gases. One specific issue on this topic is that adsorbing times are on a 50% benchmark and the absorbing times on a 90% benchmark. Are these quantitatively comparable, which is to say, do they have the same mathematical meaning to allow direct comparison across modes? In Eq. (3), it seems odd to subtract the time it takes to get to 90% in the instrument from the time it takes to get to 50% in the complete setup. Couldn't the sigmoidal fit of the adsorbing data be used to similarly estimate time it takes to get to 90% and provide a uniform comparison? On a similar note, why where timescales measured during depassivation for absorbent and passivation for adsorbent? The authors explicitly discuss that these metrics are different, so different equations are used and when they are included on the same plots a note is made. However, the authors nevertheless compare these cases, for instance stating in the abstract that "glass and uncoated and coated metals ... always caused longer delays than Teflon." I think it is important to compare across these materials, so my suggestion is not to stop comparing (e.g. removing the offending sentence in the abstract), but to put a little more care into figuring out how best to compare across categories (e.g. unify benchmarks).

Response. The decision to use a 50% benchmark for adsorptive materials was made for practical reasons. We had originally fit traces to sigmoidal curves and calculated the time to 90% as suggested by the reviewer. This approach worked well for some data sets, but not others. A number of traces, particularly at higher RH, fit very poorly to sigmoidal curves. Conversely, the 50% benchmark yields more consistent, stable results. Using the 90% benchmark for traces that do fit a sigmoidal curve well only increases the reported delay time by ~40% compared to the 50% benchmark. And since the delays caused by adsorptive materials are already larger than the 50% delay times of absorptive materials, changing to a 90% benchmark for adsorptive materials would only slightly increase this difference. So the statement that Teflon performs better does not depend on the benchmarks.

The decision to measure delays by passivation rather than depassivation was similarly made for practical reasons. Because depassivation can take a long time for some adsorbent tubing, measurement delays in adsorptive tubing were determined from passivation curves. Furthermore, because measurements of adsorptive uptake are much more common in the literature such curves may have more practical value.

To further clarify these issues we added the following text on lines 144–148:

"Although the time series are approximately sigmoidal, fitting to these types of curves resulted in poor fits for a number of experiments. The 50% benchmark used here resulted in more consistent and stable results, and although it reduced the reported delay time by ~40% compared to a sigmoidal fit, it did not significantly the comparisons between materials."

And we also added the following text on lines 126–129:

"Because adsorbent tubing can take much longer to depassivate than absorbent tubing, for these experiments measurement delays were determined from passivation curves. This approach also has more comparative value, since measurements of adsorptive uptake to various materials are much more common in the literature."

Comment 2. A lot of time is spent rationalizing and discussing the fact that the steel tubing was previously used. I'm not sure how best to handle this; in short my issue is that large sections of pages 13 and 15 discuss the potential impact of this issue on the observed results but fundamentally it is just an N of 1. If the authors truly believe that the results of steel are strongly influenced by the history of the tubing, it seems best to just leave that data out and focus on 13 instead of 14 types of tubing. Otherwise, given the amount of time these seem to need to talk about it, it is apparently a bit of an apples to oranges comparison. Maybe they could have another small paragraph if which they discuss the possibility that tubing history has an impact and present their evidence for that specific issue there.

Response. We think that it is worthwhile to note the potential importance of tubing history or gas sampling, so have added the following text on lines 287–290:

"Although some segments of the scientific community are aware of "memory effects" and the possible advantages of "conditioning" sampling lines, it is still worthwhile to raise awareness of the potentially important but unpredictable effects of tubing history on gas sampling."

*Technical Comments*

Comment 1. I actually have very few minor technical comments. I noticed no typos or specific issues in language, and believe the figures are clear and to the point.

Comment 2. I notice that the manufacturers of C-PFA also make C-FEP, and it seems to be signficantly cheaper (a little more than half the price). Given the relatively similar results between FEP and PFA, is there a reason the authors chose to test C-PFA but not C-FEP? Convincing the community to switch to conductive Teflon would be easier if cheaper, so it is a little bit unfortunate that C-FEP was not tested or discussed.

Response. When in our initial investigations we found conductive PTFE behaved similarly to PTFE we decided to seek out conductive PFA because it seemed to perform the best out of the tested materials. Based on the similarity between PFA and FEP, and the fact that the added conductive material seems to have a negligible effect on delays, we agree that C-FEP may be a much more cost-effective solution. We have added the following text on lines 350–352 to make readers aware of this:

"Conductive FEP, although not tested in this work, may combine good gas and particle transmission at approximately half the price of conductive PFA."

Comment 3. I think Figure S3 actually adds a lot of insight, and should maybe be added to the main body of the manuscript. If this change were made, the figure would need some cleanup to bring it up to the clarity standards of the current main figures.

Response. We have considered this suggestion, but decided it would not be a good use of journal space to move Figure S3 to the main body of the manuscript. The point of the figure was to show that conductive PFA and Silonite tubing efficiently pass particles, which was already expected since conductive silicone tubing is used by some investigators for particle sampling because of this property.

**Reviewer 2**

*General Comments*

In analytical chemistry of gases and particles interactions with wall material of tubing, adsorbents, and/or material of other devices are often crucial. This paper investigates the impact of various tubing materials on the recovery of 4 ketones (C6-C13) and one C12-hydrocarbon in different humidity regimes. The authors present the results clearly and well structured.

*Specific Comments*

The gas/particle phase analytical community certainly appreciates such an investigation assisting in choosing an appropriate tuning material in the applications. The tubing material selected in the experiments is well tuned far from being comprehensive. It may be worthwhile to discuss more deeply effects of humidity and aging of tubing material on adsorption/desorption processes and thus related recovery of the target compounds. The paper is recommended for publication with minor revisions considering the technical comments.

Response.

We have chosen not to expand our discussion of these issues since we think our Discussion and Conclusions already capture our thoughts and extent of understanding on aged tubing and the effects of humidity. We also note the other reviewer suggested that we delete our work on aged tubing. Our summary statements on this are given on lines 287–290:

"Although some segments of the scientific community are aware of "memory effects" and the possible advantages of "conditioning" sampling lines, it is still worthwhile to raise awareness of the potentially important but unpredictable effects of tubing history on gas sampling."

And on lines 345–348:

"If adsorbent tubing must be used, delays can be minimized by ensuring the RH is maintained above 20%. It should also be emphasized that use of adsorbent tubing can result in large memory effects and sampling artifacts, particularly upon changes in RH."

*Technical Comments*

Comment 1. The author team should thoroughly check the comma settings throughout the text. This would lead to better structured text improving text understanding.

Response. We have reviewed our manuscript and revised our use of commas as deemed necessary.

Comment 2. Page 2 line 47: . . . demonstrated that partitioning is depended . . .

Response. The reviewers have misread the sentence, which does not include this quote. On lines 47–48 it reads: "They also *demonstrated that partitioning depended* only on the saturation concentration of the organic compound and not its specific functionality."

Comment 3. Page 4/5 line 86/87: per definition ppb in not a concentration. It is a mixing ratio. It is not clear whether weight or volume of the compound of interest is used. It is recommended to use amount fraction or mol fraction throughout the text.

Response. We have changed from ppb to ppbv (volume mixing ratio) throughout the text. We also added clarifying text on lines 85–87:

"The initial concentration in the chamber was approximately 20 ppbv (mixing ratio, 1 ppbv = $2.06 \times 10^{10}$ molecules $cm^{-3}$ for the temperature and pressure of this study) for each compound prior to gas-wall partitioning."

Comment 4. Page 5 line 101: please explain how the flow of 300 ml/min was adjusted. A possible artefact formation through the flow regulating device should be discussed.

Response. The flow into the instrument was adjusted with a needle valve incorporated as part of the inlet. This did not lead to any artefacts, as noted in the text added on lines 102–104:

"The flow rate through the inlet was maintained at $0.300 \pm 0.015$ L $min^{-1}$ with a Teflon needle valve, with any delay due to absorption in the valve being accounted for as a component of the instrument delay as described below."

Comment 5. Figure 1 legend: the dashed line represents not the chamber concentration. Please describe the correct meaning of the dashed line in the legend below the figure and remove dashed line and label from the legend box in Figure 1a.

Response. We have removed the dashed line and "chamber concentration" from the legend, but have retained the dashed line in the figure. The y-axis is now labeled "Normalized Signal" instead of "Signal (Normalized to Chamber Concentration)". We have also added the following text to the caption in Figure 1:

"Signals were normalized to the values measured when sampling directly from the chamber, which were given a value of 1.0 and are represented by the dashed line in the figure."

Comment 6. Page 12 line 249: "... we hope will help inform decisions in designing sampling schemes. The meaning is not clear. Please rephrase.

Response. We have rewritten this sentence on lines 259–260 as follows:

"Instead, we discuss a few general findings that we hope will provide guidance for researchers when choosing materials for sampling lines."

Comment 7. Page 12 line 253 to line 256: "The longest measured delay times were for aluminium tubing and aluminium tubing treated with hexavalent chromate conversion coating. This treatment is intended to prevent corrosion and is used in the Potential Aerosol Mass flow reactor (Kang et al. 2007), and it does not appear to significantly affect the measurement delays." Please check for consistency: "longest measured delay times" versus "does not appear to significantly affect the measurement delays". Correct "aluminum".

Response. The chromate conversion coating does not appear to significantly alter the already long delay times of aluminum. We have clarified this by changing the text on lines 269–272 to the following:

"This coating is intended to prevent corrosion and is also used in the Potential Aerosol Mass flow reactor (Kang et al. 2007). Delays through aluminum tubing, either with or without this coating, were long, so the coating does not appear to provide and improvement in the measured tubing delays."

We also note that AMT permits either spelling of aluminum (or aluminium), as long as consistency is maintained.

Comment 8. Figure 2, legend x-axis: what is the meaning of the "*" in SIMPOL C? Please explain or delete.

Response. The caption in Figure 2 has been rewritten to clarify this as follows:

"Delay times measured for a series of 2-ketones sampled through tubing composed of different materials. Delay times were normalized to tubing length and saturation concentrations (C*) of 2-ketones were estimated using SIMPOL.1. Error bars…"

Comment 9. Page 14 line 283: Please start new sentence after "(Fig S3)". ... (Fig S3). The measurements may be . . ..

Response. We have changed the text on lines 294–296 as follows:

"Notably, Silonite tubing performed as well as PFA Teflon in terms of delays, even at low relative humidity, and exhibited good particle transmission (Fig S3). We note, however, that measurements can be influenced by humidity and VOC-VOC interactions."